# Clinicopathological Outlines of Post-COVID-19 Pulmonary Fibrosis Compared with Idiopathic Pulmonary Fibrosis

**DOI:** 10.3390/biomedicines11061739

**Published:** 2023-06-17

**Authors:** Roxana-Elena Cîrjaliu, Mariana Deacu, Ioana Gherghișan, Angela-Ștefania Marghescu, Manuela Enciu, Gabriela Izabela Băltățescu, Antonela Anca Nicolau, Doina-Ecaterina Tofolean, Oana Cristina Arghir, Ariadna-Petronela Fildan

**Affiliations:** 1Department of Pneumology, Faculty of Medicine, “Ovidius” University of Constanta, 900470 Constanta, Romania; roxana.cirjaliu@gmail.com (R.-E.C.); tofoleandoina@yahoo.com (D.-E.T.); arghir_oana@yahoo.com (O.C.A.); ariadnapetrofildan@yahoo.com (A.-P.F.); 2Clinical Emergency “St. Andrew” Hospital of Constanta, 900591 Constanta, Romania; iftimemanuela@yahoo.com; 3Department of Anatomopathology, Faculty of Medicine, “Ovidius” University of Constanta, 900470 Constanta, Romania; 4Pneumology Hospital of Constanta, 900002 Constanta, Romania; 5Department of Anatomopathology, “Carol Davila” University of Medicine and Pharmacy, 020021 Bucharest, Romania; angela.varban@drd.umfcd.ro; 6Pneumology Institute “Marius Nasta”, 50158 Bucharest, Romania; 7Center for Research and Development of the Morphological and Genetic Studies of Malignant Pathology-CEDMOG, “Ovidius” University of Constanta, 900591 Constanta, Romania; gabrielabaltatescu@yahoo.com (G.I.B.); ancanicolau@rocketmail.com (A.A.N.)

**Keywords:** post-COVID-19 pulmonary fibrosis, idiopathic pulmonary fibrosis, interstitial lung disease, pulmonary fibrosis

## Abstract

This review brings together the current knowledge regarding the risk factors and the clinical, radiologic, and histological features of both post-COVID-19 pulmonary fibrosis (PCPF) and idiopathic pulmonary fibrosis (IPF), describing the similarities and the disparities between these two diseases, using numerous databases to identify relevant articles published in English through October 2022. This review would help clinicians, pathologists, and researchers make an accurate diagnosis, which can help identify the group of patients selected for anti-fibrotic therapies and future therapeutic perspectives.

## 1. Introduction

The novel “severe acute respiratory syndrome coronavirus 2” (SARS-CoV-2) is the cause of the coronavirus disease (COVID-19), which has become a global pandemic causing millions of deaths. SARS-CoV-2 can affect various organs in the body, leading to acute organ damage and long-term sequelae [1,2]. Three years after the start of the pandemic, prospective studies regarding the long-term changes of SARS-CoV-2 infection have just begun to emerge. While clinical studies regarding the safety and effectiveness of antiviral agents and vaccines are ongoing, chronic pulmonary consequences of COVID-19 are increasingly concerning and have become more recognized.

Although pulmonary fibrosis has been observed in varying degrees of evolution in patients with SARS-CoV-2 infection, its mechanism has not been sufficiently studied and elucidated, with various causes being proposed, such as chronic inflammation and idiopathic, genetic, and age-related fibroproliferative processes. The data showed that 40% of patients with COVID-19 develop ARDS, while only 20% have severe outcomes. Pulmonary fibrosis can be a sequel of ARDS, although the radiological anomalies could be of little clinical significance and diminished after lung ventilation [3]. The prevalence of post-COVID-19 fibrosis will be visible in time, but early analysis suggests that more than a third of the recovered patients develop fibrotic anomalies. One study showed that 4% of patients with a disease duration of more than a week, 24% with a disease duration between 1 and 3 weeks, and 61% with a disease duration longer than 3 weeks develop fibrosis [4]. 

Retrospective studies have shown that over 90% of hospitalized patients have presented persistent modifications of the lungs at the moment of discharge from the hospital [5,6]. In contrast, most patients with mild forms of COVID-19 had ground glass opacities that disappeared three weeks after hospital discharge [7]. In a series of severe COVID-19 patients who had recovered, at the one-year follow-up, nearly 25% had persistent radiologic abnormalities with features characteristic of pulmonary fibrosis, including reticular opacities, septal thickening, and traction bronchiectasis [5]. Although pulmonary fibrosis is not a common complication in other forms of viral pneumonia, it might be a possible complication of COVID-19 pneumonia leading to irreversible lung dysfunction in many COVID-19 patients [8].

Pulmonary fibrosis occurs when the tissue’s restoration ability is affected by the impaired regeneration of the alveolar epithelium, with fibroblast persistence and excessive collagen deposition, leading to abnormal lung architecture [9]. Different potential contributing risk factors for persistent fibrotic lung changes in COVID-19 patients were proposed, such as age over 40 years old, hospitalization duration over 16 days, noninvasive mechanical ventilation, tachycardia at admission, acute respiratory distress syndrome (ARDS), and computer tomography (CT) score over 17 at the initial evaluation [10]. Due to the tremendous number of COVID-19 cases and the disease severity, it is crucial to consider the potential long-term implications of this disease. 

Interstitial lung diseases (ILDs) are a heterogeneous group of parenchymal lung diseases characterized by varying degrees of inflammation and fibrosis. Idiopathic pulmonary fibrosis (IPF) having an unknown origin, a chronic progressive evolution, and a poor prognosis, with a mean survival of 2.5–5 years after definite diagnosis, represents around 17–37% of diffuse interstitial lung diseases [11]. It primarily affects older adults and has exclusively pulmonary involvement, causing chronic, progressive lung scarring, as defined by the pathological histopathologic and/or radiologic pattern of usual interstitial pneumonia (UIP). The histological characteristics of the UIP pattern are marked parenchymal fibrosis with or without “honeycombing”, predominantly subpleural and paraseptal, the presence of fibroblast foci, traction bronchiectasis, and remodeling of the alveoli [12].

Regarding etiopathogenesis, current theories hypothesize that alveolar epithelial cell injury is the initiating factor and, in this sense, is essential for excluding another interstitial lung disease. The lesion consists of excessive proliferation of fibroblasts and myofibroblasts and deposition of disorganized collagen and extracellular matrix, resulting in lung architecture distortion, with or without honeycomb cyst formation [13]. 

In conclusion, PCPF and IPF determine lung fibrosis through different mechanisms, characterized by altered tissue regeneration, causing abnormal lung parenchyma. There is a need for further studies to establish the features of post-COVID-19 pulmonary fibrosis (PCPF) and its evolution, such as the permanent or progressive character that appears in other fibrotic lung diseases such as idiopathic pulmonary fibrosis (IPF).

## 2. Materials and Methods

The scope of our study is to make an accurate comparison between PCPF and IPF from etiopathological, clinical, and paraclinical points of view, to analyze the therapeutic interventions that can be used for PCPF. For our study, the eligibility criteria included the inclusion and exclusion criteria, which referred to the type of study, databases, date, language, and search terms. We selected the following studies: systematic reviews, meta-analyses, systematic literature reviews, systematic narrative reviews, systematic scoping reviews, systematic meta-review, systematic evidence reviews, systematic critical reviews, systematic integrative reviews, case reports, and case series.

We performed this literature review using the following databases: MEDLINE, Clarivate, PubMed, Scopus, Google Scholar, and Science Direct, to identify relevant articles published in English through October 2022. We used Medical Subject Headings (MeSH), standardized biomedical and health-related keywords that describe the subject of an article, as well as free text terms, to expand our search. We used the tiab (title and abstract) code after each free text term to restrict our query and find the relevant articles. The exclusion criteria included articles published in languages other than English.

Search terms included the following: COVID-19, post-COVID-19 pulmonary fibrosis, idiopathic pulmonary fibrosis, histopathologic, clinical, and imaging aspects. 

The search resulted in 2567 total articles. The most frequently used databases and the selection criteria represented the search strategy to retrieve the studies. An example is (search* [tiab] OR medline [tiab] OR clarivate [tiab] OR pubmed [tiab] OR [tiab] [tiab] OR google scholar [tiab] OR science direct [tiab]) AND (selection criteria [tiab] OR study selection [tiab] OR eligibility criteria [tiab] OR inclusion criteria [tiab] OR exclusion criteria [tiab]). 

The authors independently reviewed the titles and abstracts included in this review. The investigation was supplemented by reviewing reference lists of included studies and related review papers. 

## 3. Risk Factors

Recognizing the potential risk factors for PCPF is a primary goal for improving the clinical course of these patients. It has been demonstrated that PCPF has several shared significant risk factors with IPF (Table 1) [14]. Some studies from different countries showed that patients at higher risk for PCPF are elders, males, smokers, and patients with underlying conditions such as diabetes and cardiovascular and lung diseases [15,16,17,18]. Smoker patients with COVID-19 are 2.4 times more likely to need ICU admission and mechanical ventilation or die than nonsmokers [19,20]. Studies demonstrated that cigarette smoking determines endoplasmic reticulum stress; production of transforming growth factor beta (TGFB) [21], which mediates fibrosis; increased epithelial permeability; production of reactive oxidative species; and alteration of tissue regeneration, inducing lung micro-injuries [22].

Furthermore, different features related to the severity of the acute phase of the disease were associated with a higher risk of developing PCPF, such as the presence of dyspnea, prolonged hospital stay, intensive care unit (ICU) admission, the use of high-flow oxygen support, the need for intubation and mechanical ventilation, and development of ARDS [15,23,24]. McGroder’s study has shown that at four months of follow-up, in a population with severe COVID-19, 72% of the patients who had mechanical ventilation developed PCPF, as opposed to 20% of patients who did not [25]. Opposite to the severe forms of COVID-19, patients with mild-to-moderate disease had no fibrotic abnormalities on the CT scan follow-up [26].

As the global COVID-19 pandemic has progressed, various studies have identified different markers and biomarkers that may predict the development of PCPF [17,23]. In retrospective studies from China, higher levels of serum lactic dehydrogenase and inflammatory biomarkers, including C-reactive protein (CRP), and interleukin-6 (IL-6) were found in the subgroups of patients who presented changes compatible with pulmonary fibrosis on follow-up CT scans [17,18]. Decreased lymphocyte count [27] and lower plasma levels of interferon-γ (IFN-γ) are other laboratory markers associated with a high risk of developing PCPF [28].

Regarding the risk factors for the development of IPF, there is increasing evidence to support the role played by the intrinsic risk factors (e.g., advanced age, male sex, genetics, lung microbiome), comorbidities (e.g., diabetes mellitus, gastroesophageal reflux, obstructive sleep apnea, herpes virus infection), and extrinsic risk factors (e.g., smoking, air pollution, environmental exposures). These risk factors, some of them in common with those predisposing to PCPF, may independently increase susceptibility for IPF or work in a synergistic mode to contribute to a higher risk for disease development [29] or be associated with shorter survival rates [30]. The incidence and prevalence of IPF increase with age, being higher in adults 65 years or older. The mechanisms include altering the proliferation/apoptosis ratio [31], decreasing alveolar stability, and reducing differentiation capacity, which promotes fibrosis [32].

Around 70% of all IPF patients are male [33], males also being 1.3 times more subjected to PCPF than females [34]. Studies performed on animals gave a possible explanation, showing that female hormones have a role in protection against pulmonary fibrosis [35,36]. Another cause may be that men are more exposed to tobacco smoking and occupational exposures [33]. Smoking history increases the risk of developing IPF by 60% [37] and the risk of progression to severe forms of COVID-19 by 1.3 times [4]. The environmental and occupational factors associated with IPF are metal dust, wood dust, stone, sand, and farming substances [38].

Genetic predisposition is also a significant risk factor for PCPF and IPF (Table 2). Host genetic predisposition was proposed as a risk factor for severe courses of COVID-19 [39]. The first genome-wide association studies (GWASs) identified the 3p21.31 gene cluster (rs11385942) associated with severe forms of COVID-19 and respiratory failure and confirmed a potential involvement of the ABO blood group system [40]. The COVID-19 Host Genetics Initiative (HGI) meta-analysis also confirmed the 3p21.31 locus as significant, with two signals within the locus, one associated with severity (rs10490770) and the other with infection susceptibility (lead variant rs2271616) [41]. CXCR6 recruits CD8-resident memory T cells in the respiratory tract to combat respiratory pathogens and is a causal gene for severe disease (lead variant rs10490770) [42].

Different studies reported that several genes are associated with IPF predisposition, including the genes that encode the surfactant and proteins A and C (SFTPC, SFTPA1, SFTPA2), genes associated with telomerase dysfunction (TERT, TERC, DKC1, RTELI, PARN), genes affecting the integrity of the epithelial barrier (DSP), and genes affecting host defense (MUC5B, TOLLIP) [43,44]. The most recent meta-analysis reported an association between MUC5Brs35705950 and hospitalization due to COVID-19 [45].

The study of Fadista showed that genetic variants associated with IPF did not predispose to an increased risk of severe COVID-19 [14]. As mucins are involved in the first defense against pathogens in the airways and play an essential role in mucociliary clearance, high expression may protect against SARS-CoV-2 infection. Nevertheless, this study was driven by a single abnormal variant at the MUC5B locus, which had an apparent protective effect on the severity of COVID-19. Removal of this outlier demonstrated that the remaining variants associated with increased risk of IPF were also associated with increased risk of severe COVID-19.

Studies have shown that microbial pathogens such as Streptococcus and Staphylococcus are associated with the development and progression of IPF [46,47]. At the same time, COVID-19 patients with bacterial infections were more frequently admitted to the ICU and needed invasive ventilation [48]. A recent meta-analysis [49] showed a prevalence of bacterial co-infections of 3.5% in COVID-19 patients admitted to the ICU, with most of the cases being hospital-acquired infections with Gram-negative germs, complicating the evolution of the disease, prolonging the hospitalization time, and increasing the risk of mechanical ventilation, all of which are risk factors for PCPF development.

In conclusion, PCPF and IPF have common risk factors such as advanced age, male sex, smoker status, comorbidities, and genetic mutations. At the same time, toxic environmental exposure is a risk factor incriminated in IPF, not proven yet in PCPF.

## 4. Clinical Aspects

The clinical manifestations of PCPF and IPF are similar (Table 3), with both diseases sharing numerous symptoms, including dyspnea, dry cough, fatigue, chest pain, and weight loss, which are related to decreased life quality [50,51]. There is a lack of data for the clinical course of PCPF, but the results of the relevant prospective studies will fill this gap.

A study conducted by Farghaly assessed persistent symptoms in patients with PCPF at six-month follow-up after the acute disease and showed that the most common symptoms are dyspnea (98%), dry cough (91%), fever (70%), productive cough (19%), and chest pain (16%). The complications of PCPF that increase the risk of death are respiratory failure, sepsis, and acute kidney injury [52].

Another prospective cohort study of 76 severe COVID-19 patients requiring supplemental oxygen found a positive correlation between the presence of radiographic abnormalities (fibrotic-like patterns) 4 months after hospitalization and decreased lung function, cough, and frailty [25]. 

The study conducted by Kamal showed that almost 90% of the 287 included patients suffered from several symptoms and diseases after recovery from COVID-19. Most individuals suffered from fatigue (72.8%), anxiety (38%), joint pain (31.4%), continuous headache (28.9%), chest pain (28.9%), dementia (28.6%), depression (28.6%) and dyspnea (28.2%), while 2.4% of recovered patients were newly diagnosed with diabetes [53]. There was a positive correlation between the initial disease’s severity and post-COVID-19 manifestations, many related to the central nervous system, including continuous headache, anxiety, depression, and obsessive–compulsive disorder. 

Psychological morbidities such as anxiety and depression were also reported in IPF patients [54]. Anxiety and depression are strongly associated with health-related quality of life [55]. For IPF patients and subjects recovering from COVID-19, continuous counseling is essential to detect warning signs of developing severe manifestations and maintain good medication adherence. 

A case–control study performed on the UK population in 2017 has shown a strong correlation between dyspnea and cough as debut symptoms in IPF, appearing up to four years before the diagnosis [56].

Obstructive sleep apnea is also highly associated with IPF, with patients reporting snoring, insomnia, daytime sleepiness, and witnessed apneas. Gastro-esophageal reflux can be present in 35% to 100% of patients with IPF. These patients can be asymptomatic or present different digestive symptoms, such as regurgitation, belching, dysphagia, dysphonia, and chest pain. Cough can be associated with about 28% of cases of gastro-esophageal reflux [57]. 

The literature data have well documented the physical findings of IPF patients compared to PCPF patients. A study analyzing patients with PCPF six months after an acute episode has shown the presence of pathological auscultation sounds in 4% to 12% of the patients, represented by Velcro crackles and wheezing [58].

Fine crackles, usually in the lower posterior parts of the lung, are typically reported in IPF patients, while clubbing fingers are found in 30–50% of the cases, correlated with smooth muscle proliferation in the areas of fibrosis observed in lung biopsy. Body mass index also correlates with IPF patients’ survival [59].

In conclusion, PCPF and IPF have similar manifestations, such as dyspnea and cough, and physical findings represented by Velcro crackles, both evolving in time with respiratory failure, increasing the death risk in these patients.

## 5. Pulmonary Function Tests

Pulmonary function tests (PFTs) indicate a restrictive pattern and an altered lung diffusion capacity for carbon monoxide (DLCO) in the vast majority of cases of patients who have recovered after severe forms of COVID-19 (Table 4) [60,61]. In a recently published meta-analysis, which included 380 post-COVID-19 patients, the authors found impaired DLCO and restrictive and obstructive patterns in 39%, 15%, and 7% of subjects [61]. A high prevalence of decreased DLCO (66%) was found in patients with severe COVID-19, especially those with elevated inflammatory markers, who were more likely to develop pulmonary fibrosis [18]. The British Thoracic Society (BTS) guide suggests evaluating patients with severe COVID-19 with full PFTs 12 weeks after hospital discharge [62]. According to the findings of Cherrez-Ojeda, the impairment in PFTs appears to persist well beyond this timeframe [63]. A prospective observational study that analyzed the evolution of functional and radiological features between 3 and 6 months after hospital discharge in critical COVID-19 survivors reported the persistence of functional abnormalities such as impairments in total lung capacity (TLC) (41% and 33%) and DLCO (88% and 80%) at the end of the monitoring period [64]. Significant improvements were observed only in forced expiratory volume in 1 second (FEV1), forced vital capacity (FVC), and distance covered during the 6-minute walking test (6MWT).

Acute phase severity markers such as the presence of previous conditions (arterial hypertension and diabetes), invasive mechanical ventilation (IMV), and prone positioning during the ICU stay were associated with a low level of DLCO and its non-improvement during the 3rd and 6th months of follow-up. The decrease in DLCO could result from interstitial or pulmonary vascular abnormalities caused by critical COVID-19. The authors observed that despite the partial radiological resolution, gas–blood exchange abnormalities persisted at the six-month follow-up, which could suggest the initial establishment of an irreversible, chronic lung disorder. 

Although the restrictive pattern is consistent also in IPF, with reduced FVC and TLC in one study, 25% of the patients had normal values of TLC, and more than 50% had normal FVC (Table 4). The obstructive pattern is not typical in IPF, suggesting the coexistence of an obstructive cause. The lung function decline is a mortality predictor in IPF; as the restriction’s severity increases, FEV_1_/FVC increases, and DLco decreases. Patients with a drop bigger than 10% in FVC in 6 months or desaturate less than 88% at the 6MWT have a higher mortality risk [65]. In IPF, the decrease in DLCO usually precedes restrictive dysfunction, and DLco lower than 35% or a decline of more than 15% in one year is also associated with high mortality [66,67].

In conclusion, because the changes in forced vital capacity (FVC), total lung capacity (TLC), and DLCO are predictive factors for mortality in IPF patients, long-term monitoring through PFTs may also be justified in patients with PCPF. 

## 6. Radiologic Aspects

In the early stages of COVID-19, the most common radiological findings are bilateral “ground glass opacities” (GGOs) and consolidations, predominantly in the lower lobes, posterior and peripheral. The radiologic exam may also show a “crazy-paving” pattern, nodular opacities, halo sign, reversed halo sign, pleural effusions, cavitation, and lymph node enlargement [15,68,69]. The extent of the lesions varies significantly; they can be patchy or diffuse, and a predominance of central and upper distribution may also be present. Different pulmonary CT abnormalities may be found even in patients without respiratory symptoms. 

The lesions from the acute phase of the disease may progress to fibrotic abnormalities such as interlobular septal thickening and traction bronchiectasis, especially in survivors of critical forms of COVID-19 [70]. However, the data regarding the long-term evolution of pulmonary changes in these patients are scarce. Liu D reported that between three and four weeks after the acute COVID-19 pneumonia, a transitory extension of the GGOs occurs, with a decrease in density, an aspect described as “tinted sign”, accompanied by the distortion of the bronchovascular bundle [7]. Most patients with mild or moderate pneumonia have complete resolution of the imaged lesions in the first month after the episode; the first lesions that resolve are GGOs. Other lesions such as subpleural bands and bronchial dilations are remitted slowly, the severity of the initial clinical manifestations being the most common determinant of the resolution time. 

Regarding the severity of lung parenchymal affection, the most commonly used CT score has a range from 0 to 25, each lobe affection being visually scored on a scale of 0–5, with 0 indicating no involvement; 1, less than 5% involvement; 2, 5–25% involvement; 3, 26–49% involvement; 4, 50–75% involvement; and 5, more than 75% involvement [71]. 

Recent studies evaluated long-term longitudinal changes in chest CT findings in COVID-19 survivors. Yin X showed that GGOs remained on CT images in 30% of the subjects more than six months after discharge [72]. At the same time, reticulation was observed in 46% of cases, more commonly after severe forms of the disease, with a longer duration of hospitalization. In a one-year follow-up study, the CT scans were normal in 16% of cases, were stable in 19%, and showed a reduction in lesion extension in 65% of the patients, in reticular band opacities persisting at one-year follow-up. At the same time, the extent of GGOs was substantially reduced, and the long-term persistence of CT abnormalities after COVID-19 was correlated with the disease’s severity, high inflammatory state, and the need for more intensive ventilator requirements [73]. 

According to the CT scoring method proposed by Camiciottoli, the degree of pulmonary fibrosis was evaluated by ground glass opacity, linear opacity, interlobular septal thickening, reticulation, honeycombing, or bronchiectasis features in chest CT images [74]. In the study of Jia-Ni Zou, approximately 80% of the 284 COVID-19 patients had pulmonary fibrosis at discharge, and it was more pronounced in patients with severe disease than in those with mild/moderate form (73.8%).

Farghaly showed the presence of GGOs (95%), honeycombing (25%), and pulmonary consolidations (9%) in patients with PCPF. The CT score was higher for patients with mechanical ventilation or ICU admission. A high CT score was also associated with prolonged hospitalization and severe dyspnea [52].

PCPF changes appear in the areas where there were previously GGOs during COVID-19 pneumonia [6]. As a result, the CT distribution of the fibrotic changes will be predominantly bilateral, peripheral, and in the lower lobes. A similar distribution of CT abnormalities was also described in IPF cases, in which usual interstitial pneumonia (UIP) represents the hallmark radiologic CT pattern (Figure 1).

Regarding the high-resolution computer tomography (HRCT) findings, the presence of honeycombing is the main difference between IPF and PCPF, being essential for the definite diagnosis of IPF but rare in PCPF, with just a few cases being reported [75]. Another chest CT feature that differentiates those two diseases is the large extent of the GGOs in PCPF, while in IPF, GCOs are commonly absent or minimal or may be present in case of exacerbations. The evolution of pulmonary manifestation on CT scan is very well documented in IPF, and the data from the literature show an irreversible, progressive course [76,77], while despite the scarce data regarding the evolution of post-COVID-19 CT abnormalities, recent studies found an improvement, with a 10–40% decrease in the extent of CT lesions and no progression at 1-year follow-up CT [72,74]. Prospective studies from large cohorts undergoing more prolonged monitoring are likely further to clarify the evolution of PCPF on CT scans.

In 2011, the clinical practice guidelines for the diagnosis of IPF from the American Thoracic Society/European Respiratory Society/Japanese Respiratory Society/Latin American Thoracic Association (ATS/ERS/JRS/ALAT) proposed the concept of multidisciplinary diagnosis for IPF in patients without surgical lung biopsy (SLB) if the patient had a UIP pattern on HRCT and suggestive clinical presentation, and recommend that HRCT features of IPF should be referred to as UIP, possible UIP, or inconsistent with UIP. The imaged UIP pattern is an accurate indicator of the presence of histopathological UIP patterns. 

The UIP pattern consists of reticular opacities, honeycombing, traction bronchiectasis, and bronchiolectasis, observed with fine reticulation, and, in the case of exacerbations, GGOs. Honeycombing is a defining feature of UIP, mandatory for a definite diagnosis, associated or not with peripheral traction bronchiectasis or bronchiolectasis (Figure 2a,b).

Except for honeycombing, UIP features on the HRCT define the “possible UIP” pattern; in this case, a lung biopsy is necessary for the positive diagnosis. The presence of other abnormalities, such as extensive GGOs, micronodules, pleural abnormalities, non-honeycombing cysts, consolidation areas, or the predominance of peribronchovascular or perilymphatic distribution, is characteristic of inconsistency with the UIP pattern. However, differentiating honeycombing from traction bronchiectasis and emphysema can be difficult. Honeycombing has a thicker wall and subpleural distribution parallel with the chest’s border, while emphysema has cystic airspaces with thin walls located further away from the chest wall [78].

In conclusion, fibrotic modifications have a basal and subpleural preponderance in PCPF and IPF. Honeycombing, mandatory for the UIP pattern and diagnosis of IPF, is only found in 25% of PCPF cases, while GGOs with extensive distribution in PCPF are present only in IPF exacerbation.

## 7. Histopathologic Characterization

The histopathological data are mainly based on autopsy findings. In contrast to other viral infections such as H1N1, for which death occurred within a few days after the symptomatic debut, the patients infected with SARS-CoV-2 died within a mean duration of three weeks. An explanation may be the more progressive lung injury caused by SARS-CoV-2 that favors the repair with extracellular marker deposition. However, histopathological studies in prolonged or post-COVID-19 patients are limited because PCPF is a relatively new entity, and few studies using autopsies were performed. This is an essential factor that needs to be further considered because this lack of knowledge also limits the etiopathogenic understanding of the disease and the development of specific therapy. 

The autopsy studies show that during the course of the disease, type I and type II pneumocytes are infected by the SARS-CoV-2 virus, resulting in a cytopathic effect, pneumocyte desquamation, accumulation of fibrinoid material in alveolar spaces, and numerous inflammatory cells in the lungs, macrophages, lymphocytes, and neutrophils [79]. 

Infected type II pneumocytes contain numerous autophagosomes, ultrastructurally characterized by double membranes and the presence of organelles in the cytoplasm, also containing viral aggregates, that can be present in tracheal epithelial cells and within the extracellular mucus in the tracheal lumen. Immunohistochemical staining can demonstrate virus particles’ existence using monoclonal antibodies against the SARS-CoV-2 nucleocapsid protein [80]. All these changes represent the spectrum of diffuse alveolar damage (DAD), which is characteristic of the acute or exudative phase of severe COVID-19, with numerous reactive pneumocytes, lung hemorrhage, fibrin deposits in the alveolar spaces, interstitial edema, hyaline membranes, giant cell formation, and bronchiolitis obliterans [81,82,83,84]. Thrombotic events in pulmonary arteries may also occur in this phase.

DAD results from acute lung injury (ALI) determined by direct or indirect causes, or in the case of ARDS, requiring mechanical ventilation. ARDS is defined as acute hypoxemia with a ratio of partial pressure of arterial oxygen to the fraction of inspired oxygen (PaO_2_:FiO_2_) of a maximum of 200 mmHg. At the same time, ALI, being less severe, refers to acute hypoxemia with a PaO_2_:FiO_2_ ratio of 300 mg Hg [85]. Other histopathologic findings in the case of ARDS and ALI are acute eosinophilic pneumonia (AEP) and acute fibrinous and organizing pneumonia (AFOP) [86].

DAD has two phases: the acute phase in the first week after lung injury and the organizing or proliferative phase [87]. Two days after the lung injury, hyaline membranes are developed, while thrombi can also be seen as a result of the alteration of the coagulation without any underlying thromboembolic disorder [88]. The organizing phase is defined by cellular fibroblastic proliferation, type 2 pneumocyte hyperplasia, squamous metaplasia, and residual fibrin rest. In this phase, the hyaline membranes become integrated into the alveolar septa and cannot be seen anymore [89]. In autopsies, other modifications such as polypoid plugs, alterations of the basal membrane, and thickening of the alveolar wall were also present [82,89,90]. After this phase, DAD can be resolved gradually or evolve into interstitial fibrosis (fibrosing stage) (Figure 3a,b), with the excessive extracellular matrix, dense collagen deposition, and diffuse thickening of alveolar walls, resulting in an architectural disorder similar to other cellular and fibrous interstitial pneumonia patterns [89,91,92]. The most common finding in 30 minimal invasive autopsies was organizing DAD (70%), acute DAD (40%), and/or fibrosing patterns. Fibrosing DAD may be involved in the development of post-COVID-19 pulmonary fibrosis. There are still limited data about the pathology of prolonged disease [89]. The study performed by Hanley B on ten autopsies reported DAD presence in all the cases, as well as lymphocyte inflammation, predominantly CD4+-positive T cells, along with macrophages and scattered plasma cells. Chronic bronchiolitis was a common finding, and CD56-positive natural killer cells were rare (Figure 3b). Thromboembolism was a frequent finding in small and medium-sized vessels, without any sign of deep venous thrombosis in the external examination [93].

The histological patterns observed in organizing pneumonia are represented by fibroblasts and myofibroblasts that fill the alveolar space and ducts as an inflammatory response to the virus. This mechanism may have a favorable evolution towards resorption or may chronically evolve to excessive collagen deposition and, thus, pulmonary fibrosis [94]. In explanted lungs from patients with lung transplants due to COVID-19, the main pathologic characteristic was extensive pulmonary fibrosis, acute interstitial pneumonia or organizing pneumonia, micro-thrombosis, alveolar hemorrhage, and acute bronchopneumonia from superimposed bacterial infection [95].

The morphologic changes of the lung in COVID-19 are the following: (1) reactive epithelial changes and DAD; (2) vascular with microvascular damage, microthrombi, and acute fibrinous and organizing pneumonia; (3) fibrotic changes, with evidence of interstitial fibrosis [96]. The epithelial and vascular changes appear alone, simultaneously, or consecutively in all stages. Buja described the association of DAD with microvascular involvement and proposed three stages of lung injury: early infection stage, pulmonary stage, and severe hyper-inflammation stage. In the first stage, the morphologic aspects are of interstitial pneumonia with DAD, in some cases associated with micro-thrombosis, peripheral lung hemorrhage, and, in severe cases, pulmonary thromboembolism [97]. In COVID-19, interstitial inflammatory infiltrate is reduced (Figure 4a,b), unlike typical interstitial pneumonia.

Another study by Ackerman highlights the presence of thrombosis (Figure 5a,b), pulmonary vascular endothelitis, and angiogenetic alterations in patients with COVID-19 [98]. In contrast, Burel’s study showed the loss of pericytes, the cells responsible for micro-vessel integrity, which may trigger micro-vasculopathy [99]. Ackerman’s study has described three angiogenetic features of COVID-19, the first being severe endothelial injury associated with the destruction of the endothelial cell membranes, the second being disseminated vascular thrombosis in the lungs, associated with microangiopathy and capillaries occlusion, and the third feature describing new vessel growth in the lungs through angiogenesis [98]. One study has shown an increased number of angiotensin-converting enzyme 2 (ACE2)-positive cells in the lungs of COVID-19 patients. SARS-CoV-2 within the endothelial cells suggests perivascular inflammation, the direct effects of the virus causing endothelial injury [100].

Some authors suggest two patterns of fatal COVID-19, with different clinical courses: one with high viral load and high cytokine expression in the lung but a limited morphologic expression, and a second one with low viral load and cytokine expression but a large number of immune cells (including CD8 + T lymphocytes and macrophages), which correlates with the presence of DAD [101].

DAD’s proliferative/organizing phase shows type II pneumocyte hyperplasia, reactive pneumocytes (Figure 6a), alveolar wall thickening, and myofibroblast proliferation (Figure 6b). A case report of an 80-year-old woman with subsequent negative SARS-CoV-2 at the time of the autopsy showed severe reactive and inflammatory changes in all the lung samples. The architecture was destroyed in larger areas with fibrinous organization and collagenized fibrosis. Widespread angiogenesis was seen, along with focal bleeding. Local moderate chronic inflammation dominated by lymphocytes was present. The fibrosis had a honeycomb-like pattern with enlarged airspaces and bronchial metaplasia in some areas. A small subpleural area showed alveoli with hyaline membranes representing the acute stage of lung injury, as is seen in acute DAD.

The guideline panel updated the diagnostic criteria for idiopathic lung fibrosis and defined patterns of UIP, probable UIP, indeterminate UIP, and alternate diagnosis. To confirm those cases, they recommended performing bronchoalveolar lavage (BAL) and surgical lung biopsy. 

The 2018 ATS/ERS/JRS/ALAT guideline classifies the histopathological findings into UIP, probable UIP, and indeterminate UIP. IPF is macroscopically characterized by honeycombing, consisting of fibrosis in the interior part of the lobes, predominantly subpleural. The microscopic aspect of the UIP pattern is composed of patchy dense fibrosis, with architectural distortion, predominantly in the periphery of the lobule and paraseptal, with a regular central portion of the lobule. As a result, the patterns will evolve from chronic to acute to absent from the periphery to the centum of the lobule. The honeycombing aspect can be seen subpleural as dense fibrosis surrounding bronchial epithelium, which lines irregular airspaces in the centrum of the lobule, where significant inflammation and fibrosis are absent. Fibroblast foci can be seen in the area between regular regions of the lobule and fibrotic lesions, arranged parallel with the alveolar foci, on a basophilic myxoid ground (Figure 7a,b).

Some histological UIP features characterize a probable UIP pattern to the extent that precludes the definitive diagnosis, the absence of elements of an alternative diagnosis, or the presence of honeycombing alone. Indeterminate UIP is characterized by fibrosis with or without architectural distortion, suggesting UIP secondary to other causes or non-suggestive for UIP or features of UIP, along with elements of an alternative diagnosis. An alternative diagnosis is indicated by histological aspects suggestive of other diseases at biopsy, such as airway-centered lesions, granulomas, interstitial inflammation without fibrosis, chronic fibrous pleuritis, and hyaline membranes [102].

The histopathological guidelines for UIP were changed in 2021 and require advanced fibrosis with distortion of the architecture, beginning at the periphery of the lobule and going to the centrilobular regions. At this level, fibroblast foci are often encountered as evidence of active injury, usually situated at the interface between fibrotic and non-fibrotic areas, without features characteristic for an alternative diagnosis. A lack of inflammation and ununiform affection of lung parenchyma characterize UIP.

In acute exacerbations of UIP, an ALI pattern can be seen, predominantly in the regions without chronic fibrosis, while the peripheric fibrotic lesions are unmodified. In the centrum of the lobule, we can see modifications of DAD, such as type II pneumocyte hyperplasia, diffuse alveolar septal thickening caused by edema, accumulation of airspace fibrin or nonspecific changes such as thrombi within small pulmonary arteries, distal airway squamous metaplasia, and the presence of hyaline membranes [78].

In acute exacerbation of IPF, we can find DAD and UIP features, so DAD is a histopathological feature in IPF and COVID-19. Numerous factors, such as infections, shock, sepsis, connective tissue disorders, and disseminated intravascular coagulation, can cause DAD. When the etiology is unidentified, it is called acute interstitial pneumonia, previously referred to as Hamman–Rich syndrome [103].

In conclusion, IPF and PCPF both cause DAD and lung fibrosis. The data regarding PCPF are currently limited, and further studies with histopathological examination are needed.

## 8. Therapeutic Perspectives

In COVID-19, cytokine storms, oxidative stress, and inflammation are involved, so the proposed therapy for PCPF consists of anti-fibrotic and anti-inflammatory drugs.

Pirfenidone (a pyridine) and nintedanib (a tyrosine kinase inhibitor) are anti-fibrotic drugs used in IPF that reduce lung function decline by 50% and improve life expectancy by 2–5 years [104]. Neither of these drugs has an immunosuppressive effect, so they should not be stopped in case of infections.

Since April 2020, these drugs have been available exclusively in oral form. As a result, it is impossible to administrate them in the case of mechanically ventilated and intubated patients, such as patients with severe COVID-19. A form of pirfenidone with inhalator administration is under evaluation for COVID-19 patients. Pirfenidone should not be administrated in patients with a glomerular filtration rate of less than 30 mL/min per 1·73 m^2^. Patients with severe COVID-19 are at high risk of developing renal dysfunction, so pirfenidone should be carefully considered in these patients. 

Pirfenidone and nintedanib can determine hepatotoxicity, while liver function test alterations are commonly found in severe COVID-19, so in this case, temporary disruption of the anti-fibrotic treatment might be necessary [105].

Nintedanib has an increased risk of bleeding in case of concomitant administration with a full dose of an anticoagulant. COVID-19 patients have an increased risk of acute pulmonary embolism, and anticoagulant therapy is necessary. Anti-fibrotic treatment for COVID-19 might be helpful to patients with a poor prognosis and a high risk of developing pulmonary fibrosis and ALI [106]. 

The SARS-CoV-2 spike protein has an Arg-Gly-Asp integrin-binding domain raising the possibility that inhibitory integrin or galectin therapies might be helpful as COVID-19 treatment. Some drugs in development can target molecules from TGF-β pathways, such as those against αvβ6 integrin (BG0001, PLN-74809) and galectins (TD139). Studies on mice have shown that those who did not have αvβ6 integrin or had treatment with an αvβ6 blocking antibody had increased protection against viral infections [107]. In another study on mice, galectin Gal-3 determined reduced pulmonary inflammation and protection against TGF-β-induced lung injury and fibrosis [108].

mTOR is a target in IPF, and two recent studies showed that mTOR might be an anti-SARS-CoV-2 target, with rapamycin being considered in COVID-19 patients [109]. 

Pentraxins are response proteins of the acute phase, with a role in immunity and inflammation. PRM-151, an analog of SAP (PTX2), has shown promising results in IPF trials. SAP determines suppression of JNK family signaling, as a JNK1 inhibitor preventing fibrosis 55 and inhibiting ALI [110]. 

C21 role (an agonist of AT2R) is studied for COVID-19 and has clinical trial applicability in IPF, having anti-inflammatory properties. ACE2 receptors are the primary SAR-CoV-2 receptors. A study had shown that in patients who took AT1R blockers before hospitalization, the risk of severe COVID-19 was significantly decreased [111]. 

Treamid or bisamide derivative of dicarboxylic acid (BDDA) is an experimental drug with promising results used in animals with pulmonary fibrosis that inhibits the production and deposition of collagen, being in trial for use in cases of IPF and post-COVID-19 fibrosis [112].

LYT-100 (deupirfenidon) is an N-aryl-pyridone derivative, an analog of pirfenidone, which has an anti-fibrotic effect and is in trial for use in cases of COVID-19 and IPF [113].

Corticosteroids can be used in COVID-19 and IPF exacerbation. Long-term use of corticoid therapy might reduce the risk or severity of PCPF in rats, with IPF slowing down fibrosis progression [114].

An alternative therapy that might also prevent lung fibrosis is azithromycin, a broad-spectrum macrolide, which has also proved to have antiviral and immunomodulatory effects. For these reasons, it was suggested that this potential therapy for COVID-19 can reduce the risk of PCPF [115].

Histone deacetylase inhibitors can be an alternative therapy in PCPF, with increased activity of histone deacetylase promoting the activity of TGF-ß, leading to collagen synthesis and fibrosis. 

Biochanin A (isoflavone) is believed to target TGF-ß-induced fibrosis and can be an alternative therapy considered in PCPF which needs further study. A study showed a significant decrease in TGF-ß expression and collagen deposition in the lungs of the mice treated with Biochanin A [116].

Pulmonary rehabilitation, which includes exercise training, education, and behavioral changes, can improve physical and psychological conditions in cases of pulmonary fibrosis [117]. 

On the other hand, oxygen therapy is beneficial in IPF in patients with severe oxygen desaturation during exercise or resting hypoxemia, improving their symptoms and quality of life. Consequently, oxygen therapy is also an essential part of PCPF treatment [118].

In conclusion, there is a lack of effective specific treatment for PCPF, with numerous drug trials being studied; two are represented by nintedanib and pirfenidone, which are used for IPF and also have promising results in the treatment of PCPF.

## 9. Conclusions

There are numerous similar features between PCPF and IPF regarding the clinical aspects, risk factors, pulmonary function tests, and imaging and histopathological aspects, as well as some notable differences. As for clinical aspects, both pathologies have common symptoms such as dyspnea and cough; functionally, restriction syndrome and a decrease in DLco can be observed; radiologically, we can find signs of fibrosis in both cases, the main lesions being GGOs in PCPF, while the aspect is defined as UIP in IPF. Regarding histopathology, findings of lung fibrosis in COVID-19 are limited, and PCPF causes significant, irreversible consequences that affect patients’ quality of life after SARS-CoV-2 infection. Further studies with histopathological examination are needed. DAD appears in PCPF but can also occur in an IPF exacerbation alongside UIP features. More studies need to be done to determine specific effective PCPF therapy. Currently, nintedanib and pirfenidone used for IPF treatment are studied as anti-fibrotic agents for PCPF.

This review would help clinicians, pathologists, and researchers better understand the mechanisms of fibrosis, make a diagnosis as accurate as possible, and help identify the patients who can be selected for anti-fibrotic therapies and future therapeutic perspectives.

## Figures and Tables

**Figure 1 biomedicines-11-01739-f001:**
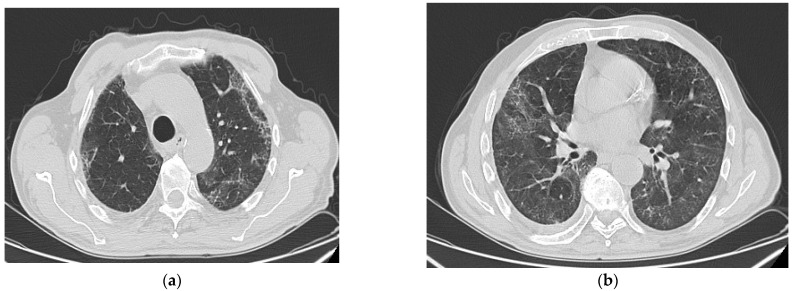
*Courtesy of Dr Oana Cristina Arghir, who provided images of post-COVID-19 pulmonary fibrosis from Pneumology Hospital of Constanta, Romania*. Thoracic CT scan of a patient with PCF, one year after the acute episode, showing subpleural and peribronchovascular reticular opacities, traction bronchiectasis (**a**), and GGO bilateral and fibrotic lines predominantly in the upper lobes (**b**), in a 77-year-old male, with history of severe COVID-19 in October 2021 and PCPF in October 2022.

**Figure 2 biomedicines-11-01739-f002:**
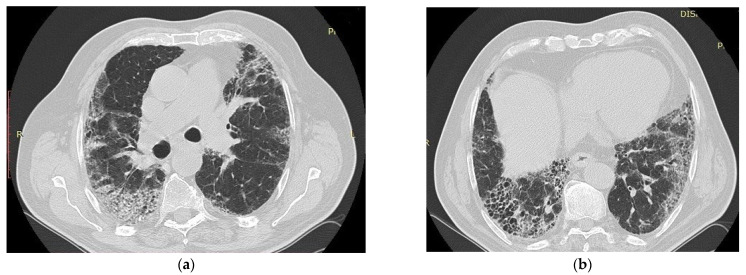
*Courtesy of Dr Ariadna Petronela Fildan, who provided images of IPF from Pneumology Hospital of Constanta, Romania*. Thoracic CT scan of a patient with IPF showing honeycombing and reticular opacities with the basal and subpleural distribution in a 73-year-old male smoker, with exposure to environmental pollutants (**a**), with idiopathic pulmonary fibrosis (**b**), who died in 2017, 3 years after the diagnosis.

**Figure 3 biomedicines-11-01739-f003:**
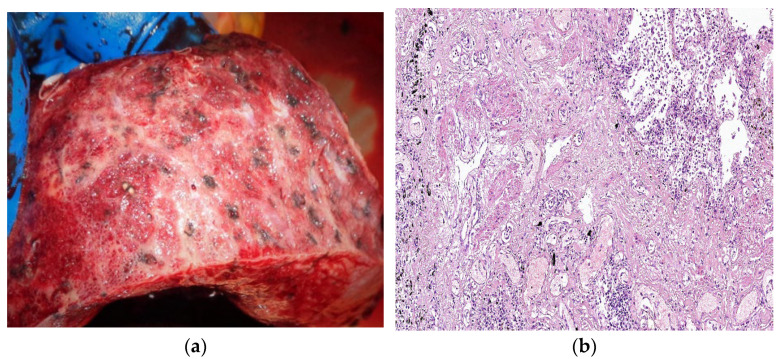
*Courtesy of Dr Mariana Deacu, who provided images of post-COVID-19 pulmonary fibrosis from “St. Andrew” Emergency County Hospital of Constanta, Romania*. (**a**) Macroscopic and (**b**) microscopic aspect of the lung in a 55-year-old male patient with COVID-19 interstitial pneumonia in the fibrosing stage, whose autopsy was performed three months after SARS-CoV-2 infection, showing architectural disorder caused by excessive extracellular matrix, dense collagen deposition, and diffuse thickening of alveolar walls. HE ×40.

**Figure 4 biomedicines-11-01739-f004:**
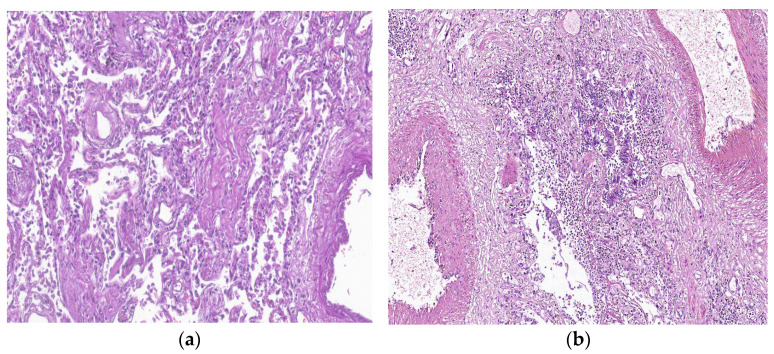
*Courtesy of Dr Mariana Deacu, who provided images of post-COVID-19 pulmonary fibrosis from “St. Andrew” Emergency County Hospital of Constanta, Romania*. Microscopic view of the lung in a 68-year-old patient with COVID-19, whose necropsy was performed five weeks after SARS-CoV-2 infection, showing (**a**) interstitial inflammatory infiltrate reduced and (**b**) chronic bronchiolitis. HE ×100.

**Figure 5 biomedicines-11-01739-f005:**
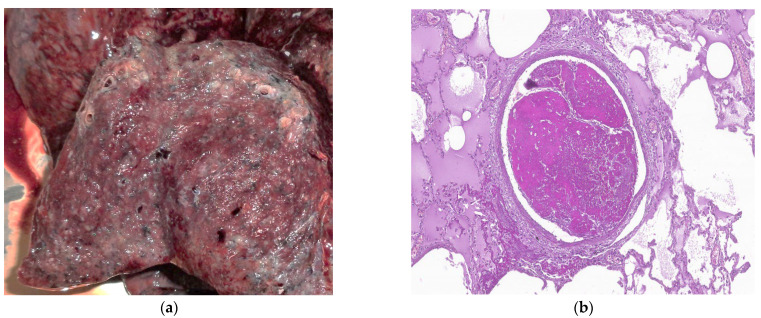
*Courtesy of Dr Mariana Deacu, who provided images of post-COVID-19 pulmonary fibrosis from “St. Andrew” Emergency County Hospital of Constanta, Romania*. (**a**) Macroscopic and (**b**) microscopic aspects of the lung in a 62-year-old patient showing thrombosis, whose necropsy was performed three weeks after SARS-CoV-2 infection, showing pulmonary vascular endothelitis and angiogenetic alterations. HE ×40.

**Figure 6 biomedicines-11-01739-f006:**
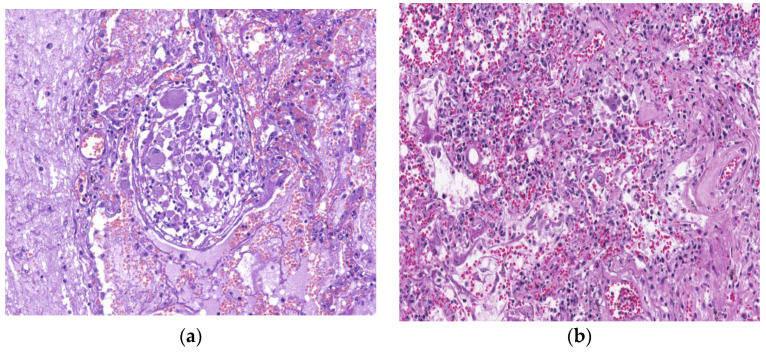
*Courtesy of Dr Mariana Deacu, who provided images of post-COVID-19 pulmonary fibrosis from “St. Andrew” Emergency County Hospital of Constanta, Romania*. Microscopic view of the lung in a 68-year-old patient, whose necropsy was performed eight months after SARS-CoV-2 infection, showing (**a**) type II pneumocyte hyperplasia and reactive pneumocytes, (**b**) alveolar wall thickening, and myofibroblast proliferation. HE ×100.

**Figure 7 biomedicines-11-01739-f007:**
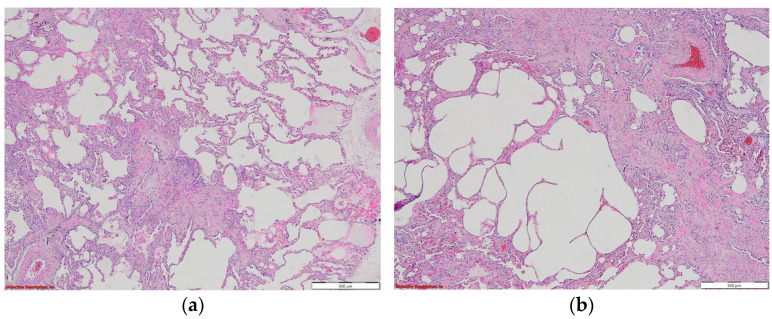
*Courtesy of Dr Angela-Ștefania Marghescu, who provided images of IPF from Pneumology Institute “Marius Nasta” Bucharest, Romania*. (**a**) Spatial variability, with normal lung parenchyma mixed with fibrotic areas (fibroblastic focus); HE ×40. (**b**) Interstitial fibrosis with architectural distortion and cystic changes in the pulmonary parenchyma; HE ×40, in a 62-year-old woman, whose necropsy was performed three months after SARS-CoV-2 infection, with marked pulmonary fibrosis.

**Table 1 biomedicines-11-01739-t001:** Frequent risk factors for PCPF and IPF.

Risk Factor	PCPF	IPF
Gender-male	yes	yes
Advanced age	yes	yes
Smoking	yes	yes
Comorbidities	yes	yes
Toxic environmental exposure	no	yes
Severity of dyspnea	yes	yes
Prolonged hospital stay	yes	no
ICU admission	yes	yes
Mechanical ventilation	yes	yes
Lung microbiome	yes	yes

**Table 2 biomedicines-11-01739-t002:** Genetic mutations in IPF and PCPF.

Gene	Mutation Consequence
*SPTPC*	Altered encoding of surfactant proteins
*SPTPA1*	Altered encoding of surfactant proteins
*SFTPA2*	Altered encoding of surfactant proteins
*TERT*	Telomere shortening
*TERC*	Telomere shortening
*DKC1*	Telomere shortening
*RTELI*	Telomere shortening
*PARN*	Telomere shortening
*DSP*	The affection of the epithelial barrier
*MUC5B*	Host defense affection
*TOLLIP*	Host defense affection

**Table 3 biomedicines-11-01739-t003:** Clinical aspects in PCPF and IPF.

Clinical Aspect	PCPF	IPF
Dyspnea	yes	yes
Cough	yes	yes
Fever	yes	no
Fatigue	yes	yes
Chest pain	yes	yes
Depression	yes	yes
Velcro crackles	yes	yes
Wheezing	yes	no
Clubbing fingers	no	yes

**Table 4 biomedicines-11-01739-t004:** PFTs in PCPF and IPF.

PFT	PCPF	IPF
Restrictive dysfunction	yes	yes
Obstructive dysfunction	yes	Coexistence of obstructive disease
Decreased DLCO	yes	yes
6MWT desaturation	yes	yes

## Data Availability

Not applicable.

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
