# Peer review of "Clinicopathological Outlines of Post-COVID-19 Pulmonary Fibrosis Compared with Idiopathic Pulmonary Fibrosis"

_biomedicines, 2023, doi:10.3390/biomedicines11061739_

Round 1

Reviewer 1 Report

The review paper combines the most recent knowledge on post-COVID-19 interstitial pulmonary fibrosis (PCPF) and Idiopathic Pulmonary Fibrosis (IPF). This collection of data can provide a comprehensive understanding of the two diseases.

Comments:

1. Materials and methods: The number of articles resulting from the search (2,567 total articles) indicates a thorough search process. However, it could be helpful to include more information about how the search terms were put together and filtered, as well as any particular inclusion or exclusion criteria used.

2. Risk factors: Without addressing any confounding factors or biases that might affect the findings, the review identifies risk factors frequently linked to severe COVID-19 and IPF. To establish a thorough understanding of the relationship between risk factors and PCPF, it is crucial to take into account and manage such issues.

3. Pulmonary function tests: The review indicates that post-COVID patients frequently have impaired DLCO, restrictive patterns, and obstructive patterns but does not give precise percentages or any other pertinent information.

4. Radiologic aspects: The review focuses more on the radiologic aspects of IPF rather than providing a comprehensive comparison with post-COVID pulmonary fibrosis. The discussion on post-COVID pulmonary fibrosis is relatively brief and lacks in-depth analysis and specific radiologic comparisons.

5. Histopathologic characterization: The review notes that there have been few histological studies in extended or post-COVID-19 patients, but it offers no additional information or justifications. It would be helpful to expand on this limitation and discuss the implications of limited data for understanding the histopathological characteristics of post-COVID-19 pulmonary fibrosis.

6.  Therapeutic perspectives: The review mainly focuses on the potential therapeutic perspectives for post-COVID pulmonary fibrosis and IPF, however, it does not provide a balanced discussion of potential limitations, controversies, or alternative strategies. Including a more comprehensive analysis would strengthen the overall argument.

7. Conclusions: The conclusion does not include the treatment options for PCPF and IPF.

Author Response

  1. Materials and methods: The number of articles resulting from the search (2,567 total articles) indicates a thorough search process. However, it could be helpful to include more information about how the search terms were put together and filtered, as well as any particular inclusion or exclusion criteria used.

            Thank you very much for your comments and suggestions. I extended the material and methods section as presented in lines 92-117.

  1. Risk factors: Without addressing any confounding factors or biases that might affect the findings, the review identifies risk factors frequently linked to severe COVID-19 and IPF. To establish a thorough understanding of the relationship between risk factors and PCPF, it is crucial to take into account and manage such issues.

Thank you very much for your comments and suggestions. I modified the chapter referring to PCPF and IPF risk factors.

  1. Pulmonary function tests: The review indicates that post-COVID patients frequently have impaired DLCO, restrictive patterns, and obstructive patterns but does not give precise percentages or any other pertinent information.

Thank you very much for your comments and suggestions. I extended this chapter and give pertinent studies as well as precise percentages.

  1. Radiologic aspects: The review focuses more on the radiologic aspects of IPF rather than providing a comprehensive comparison with post-COVID pulmonary fibrosis. The discussion on post-COVID pulmonary fibrosis is relatively brief and lacks in-depth analysis and specific radiologic comparisons.

Thank you very much for your comments and suggestions. I added data regarding PCPF radiologic chapter and a comparison between IPF and PCPF.

  1. Histopathologic characterization: The review notes that there have been few histological studies in extended or post-COVID-19 patients, but it offers no additional information or justifications. It would be helpful to expand on this limitation and discuss the implications of limited data for understanding the histopathological characteristics of post-COVID-19 pulmonary fibrosis.

Thank you very much for your comments and suggestions. I added data regarding PCPF radiologic chapter and a comparison between IPF and PCPF.

  1.  Therapeutic perspectives: The review mainly focuses on the potential therapeutic perspectives for post-COVID pulmonary fibrosis and IPF, however, it does not provide a balanced discussion of potential limitations, controversies, or alternative strategies. Including a more comprehensive analysis would strengthen the overall argument.

Thank you very much for your comments and suggestions. I added data regarding therapeutic measures in IPF and PCPF.

  1. Conclusions: The conclusion does not include the treatment options for PCPF and IPF.

Thank you very much for your comments and suggestions. I added data regardingtreatment in the conclusion.

Reviewer 2 Report

This review article provides a summary of the clinic-pathology differences between post-COVID-19 associated lung fibrosis and idiopathic pulmonary fibrosis (IPF). Although the review is comprehensive, there are areas that could be improved.

1.     To enhance readability and facilitate understanding for readers, it would be beneficial to present the parameters in a tabular format. This would strengthen the manuscript and make it easier for readers to grasp the information.

2.     The review could be more focused on the specific relationship between IPF and COVID-19. For instance, while the discussion of psychological aspects is relevant to chronic respiratory diseases, it may be more appropriate to emphasize their connection to IPF and COVID-19 specifically. Additionally, some assumptions were made in the review. For example, Reference #50 suggested a potential link between bacterial co-infection among ICU COVID-19 cases and fibrosis complications, but this study did not directly establish that connection.

3.     It would be valuable to highlight sections that explore potential therapeutic interventions during the early and later phases of COVID-19 infection, aimed at preventing or stabilizing lung fibrosis. By emphasizing these sections, the review can focus on aspects that may contribute to the understanding of pathogenesis.

4.     To increase clarity and conciseness, adding a short summarization paragraph at the end of each section would be helpful. This would provide a brief overview of the key points discussed, considering the lengthy nature of the review.

By implementing these suggestions, the revised version of the review article can be further improved in terms of readability, focus, and clarity for the readers.

The manuscript can be condensed by rewording and eliminating redundant sentences to make it more concise.

Author Response

  1. 1.To enhance readability and facilitate understanding for readers, it would be beneficial to present the parameters in a tabular format. This would strengthen the manuscript and make it easier for readers to grasp the information.

Thank you very much for your comments and suggestions. I added tables regarding the results.

  1. The review could be more focused on the specific relationship between IPF and COVID-19. For instance, while the discussion of psychological aspects is relevant to chronic respiratory diseases, it may be more appropriate to emphasize their connection to IPF and COVID-19 specifically. Additionally, some assumptions were made in the review. For example, Reference #50 suggested a potential link between bacterial co-infection among ICU COVID-19 cases and fibrosis complications, but this study did not directly establish that connection.

Thank you very much for your comments and suggestions. I added data referring to IPF and PCPF instead of focusing on COVID-19 and i corrected reference 50.

  1. It would be valuable to highlight sections that explore potential therapeutic interventions during the early and later phases of COVID-19 infection, aimed at preventing or stabilizing lung fibrosis. By emphasizing these sections, the review can focus on aspects that may contribute to the understanding of pathogenesis.

Thank you very much for your comments and suggestions. I added more data concerning to therapeutic suggestions.

  1. To increase clarity and conciseness, adding a short summarization paragraph at the end of each section would be helpful. This would provide a brief overview of the key points discussed, considering the lengthy nature of the review.

Thank you very much for your comments and suggestions. I added a short summarization at the end of each section.

Comments on the Quality of English Language: The manuscript can be condensed by rewording and eliminating redundant sentences to make it more concise.

Thank you very much for your comments and suggestions. I reworded the manuscript and eliminated redundant sentences.

Round 2

Reviewer 1 Report

No more comments.

Author Response

Thank you!

Reviewer 2 Report

The authors have extensively revised the manuscript. A few points to be further reviewed and considered to further modify.

1. The current title states, "Pathology outline.....". The review included multiple clinical parameters. I would consider changing the title to "Clinicopathological outline..." or something reflecting clinical data in the manuscript.

2. Adding the timing post-COVID-19 in all histopathology figures would be more informative. The chronicity after COVID-19 infection and the time to develop lung fibrosis (when histopathology was taken) provided insight to the underlying pathogenesis. For instance, DAD in COVID-19 resembled acute exacerbation of IPF, as the authors elaborated. COVID-19 undergoing lung transplantation generally had NSIP pattern whereas UIP pattern is typical for IPF. 

Fair. Further editing may be required.

Author Response

  1. The current title states, "Pathology outline.....". The review included multiple clinical parameters. I would consider changing the title to "Clinicopathological outline..." or something reflecting clinical data in the manuscript.

Thank you for your comment. I have modified the title.

2. Adding the timing post-COVID-19 in all histopathology figures would be more informative. The chronicity after COVID-19 infection and the time to develop lung fibrosis (when histopathology was taken) provided insight to the underlying pathogenesis. For instance, DAD in COVID-19 resembled acute exacerbation of IPF, as the authors elaborated. COVID-19 undergoing lung transplantation generally had NSIP pattern whereas UIP pattern is typical for IPF. 

Thank you for your comment. I added the timing  in all histopathology figures